# Metamaterial-enabled asymmetric negative refraction of GHz mechanical waves

Simone Zanotto [1]✉, Giorgio Biasiol [2], Paulo V. Santos [3] & Alessandro Pitanti [1]

Wave refraction at an interface between different materials is a basic yet fundamental phenomenon, transversal to several scientific realms – electromagnetism, gas and fluid acoustics, solid mechanics, and possibly also matter waves. Under specific circumstances, mostly enabled by structuration below the wavelength scale, i.e., through the metamaterial approach, waves undergo negative refraction, eventually enabling superlensing and transformation optics. However, presently known negative refraction systems are symmetric, in that they cannot distinguish between positive and negative angles of incidence. Exploiting a metamaterial with an asymmetric unit cell, we demonstrate that the aforementioned symmetry can be broken, ultimately relying on the specific shape of the Bloch mode isofrequency curves. Our study specialized upon a mechanical metamaterial operating at GHz frequency, which is by itself a building block for advanced technologies such as chip-scale hybrid optomechanical and electromechanical devices. However, the phenomenon is based on general wave theory concepts, and it applies to any frequency and time scale for any kind of linear waves, provided that a suitable shaping of the isofrequency contours is implemented.

Coherent phenomena in wave systems strongly depend on the underlying symmetries of the potential energy landscape. Noether's theorem[1] and its extensions link together symmetries and conserved physical quantities, ultimately allowing scientists to tailor specific effects by simply acting–through advanced machining and microfabrication techniques–on the system's geometrical design. In photonics this has led to various effects ranging from Bloch waves and band-gap opening[2] to far-field shaping[3], enabling, more recently, exotic features connected to the system topology[4,5]. Simple geometrical design can also lead to topologically protected transport through helical channels[6], via the formation of artificial, time-reversal preserving gauge fields[7,8]. These concepts, whose natural implementation on the electron wavefunction can be found in particular solid-state crystals, or in artificially created metamaterials for photons, are starting to be explored in the realms of acoustic and phononics[9,10]. Recent advances in the generation and control of mechanical waves with

frequencies spanning from the Hz to the sub-THz ranges has pushed the research community to the investigation of symmetry-based effects for routing, localization, and manipulation of vibrational waves[11,12]. Most of the efforts have been directed to the acoustic domain (i.e., ~1 Hz–10s of kHz), where complex, 3D metamaterials can be fabricated by macroscopic processes such as 3D printing[13–16]. Among the various results, beam squeezing[17], phonon routing[18] and negative refraction[19] have been recently reported. Much fewer experimental works have focused instead on the microscopic chip-scale, where phonon-based manipulation can be useful for hybrid devices for quantum applications[20–22]. To date, a large number of theoretical proposals have suggested the possibility of obtaining various topological effects in the 1–10 GHz range with particular designs in Si[23,24], GaAs[25] or by using external magnetic fields[26]. However, the highest frequency experimental realization of an on-chip mechanical device showing topological features is of only ~15 MHz [ref. 27]. It is

[1]NEST, Istituto Nanoscienze-CNR and Scuola Normale Superiore, Piazza San Silvestro 12, 56127 Pisa, Italy. [2]Istituto Officina dei Materiali CNR, Laboratorio TASC, 34149 Trieste, Italy. [3]Paul-Drude-Institut für Festkörperelektronik, Leibniz-Institut im Forschungsverbund Berlin e. V., 5-7, Hausvogteiplatz, Berlin 10117, Germany. ✉e-mail: simone.zanotto@nano.cnr.it

worth mentioning that topological effects have been investigated in one-dimensional heterostructures up to 300 GHz [ref. 28], yet this particular, non-planar configuration is not ideal for integration with other on-chip technologies. A full control of the symmetries on the chip-scale and at GHz frequency would unveil powerful, topological effects, which can revolutionize solid-state devices. Asymmetric and non-reciprocal phonon transport could be used for improved thermal shielding and topologically protected waveguiding, with relevance for superconducting microwave quantum technologies, high-fidelity qubit-qubit on-chip interconnections[29], as well as for ultrasensitive, sub-Kelvin temperature and heat sensing[30].

In this Article, we report on GHz mechanical wave control by GaAs metamaterials where the unit cell lacks any kind of in-plane geometrical symmetry, resulting in a strong asymmetry of Bloch modes. Using scanning laser interferometry, we map the reciprocal space at selected frequencies using narrow-band surface acoustic waves (SAW) as excitation. Relying on the extracted isofrequency contours and on numerical simulations, we identified peculiar scattering configurations, including the not-yet-observed joint presence of asymmetric refraction and negative refraction[31–37]. The powerful flexibility by design of our structures will eventually allow for accessing multiple configurations for phonon routing within the very same device.

## Results

### Mechanism of asymmetric negative refraction

We consider refraction at the interface between an unpatterned and a patterned region defined within a mechanically compliant slab, i.e., between an ordinary mechanical material and an anisotropic, asymmetric metamaterial [see Fig. 1a, b]. A mechanical wave impinging at an angle of incidence $\theta_i$ is refracted into a wave with an angle of refraction $\theta_r$. Ordinary (positive) refraction occurs when both $\theta_i$ and $\theta_r$ have the same sign; negative refraction instead connects angles of opposite sign −in other words, it redirects the beam towards the same side of the axis normal to the interface (dashed line in Fig. 1a). In addition to the positive or negative character of refraction, another aspect to be considered is the behavior upon a change in sign of the incident angle: depending on this, one can speak of symmetric or asymmetric

refraction, which we define below. In the case of symmetric refraction, when the incident angle is changed in sign, the refracted angle also changes sign, keeping the same magnitude. Conversely, in the case of asymmetric refraction, when the incident angle is changed in sign, the new refracted angle does not maintain any relation with the former one. In summary,

$$\text{symmetric refraction: } \theta_i \to -\theta_i \Rightarrow \theta_r \to -\theta_r$$
$$\text{asymmetric refraction: } \theta_i \to -\theta_i \Rightarrow \theta_r \to \theta_r'$$

Intriguingly, our phononic structure allows for the observation of *asymmetric negative refraction*, as sketched in Fig. 1a, c, d and Supplementary Fig. 1. Switching the incident angle sign, it is possible to go from ordinary (red arrows) to negative refraction (blue arrows). This effect can be understood by looking at the qualitative sketch of the isofrequency curves in the reciprocal space reported in (c), where the energy-increasing trend of the isolines has been color-coded from pink to violet. Here, $f_0$ is a frequency illustratively around 1.18 GHz; quantitative data are shown below. The group velocity of a mechanical wave[38] is proportional to the gradient of its 2D frequency dispersion $f(k_x, k_y)$:

$$\mathbf{v}_g = 2\pi \nabla_\mathbf{k} f(k_x, k_y) \tag{1}$$

Let's start by considering two mechanical waves propagating as the antisymmetric Lamb mode on an unpatterned slab with wavevector components $k_x$ and $-k_x$, respectively. For an appropriate choice of the GaAs crystal orientation, these correspond to opposite incident angles, as indicated by (i) and (i') in Fig. 1d, respectively. Thanks to the discrete translational invariance, and assuming an infinitely extended sample, the wavevector component parallel to the interface between the unpatterned and patterned regions is conserved, resulting in the selective excitation of the (ii) and (ii') modes in panel (c), respectively. No other modes are involved in the refraction process, since their $\mathbf{v}_g$ would have a negative $\hat{\mathbf{y}}$ component (backward propagation). The asymmetric shape of the isofrequency contours, that depends upon

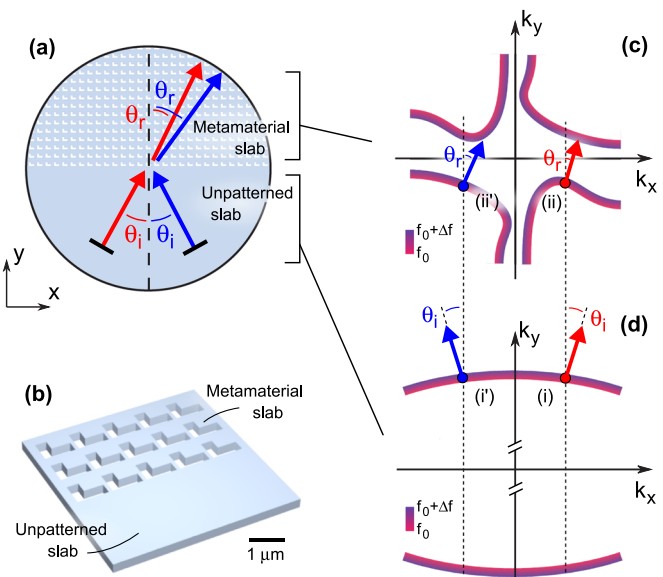
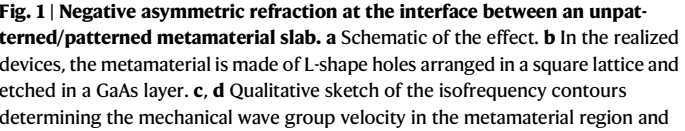
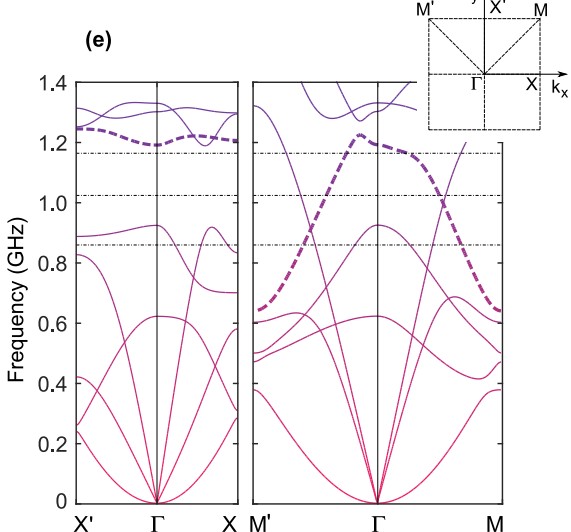

**Fig. 1 | Negative asymmetric refraction at the interface between an unpatterned/patterned metamaterial slab. a** Schematic of the effect. **b** In the realized devices, the metamaterial is made of L-shape holes arranged in a square lattice and etched in a GaAs layer. **c, d** Qualitative sketch of the isofrequency contours determining the mechanical wave group velocity in the metamaterial region and homogeneous slab. **e** Quantitative Bloch mode dispersion along high-symmetry paths in the first Brillouin zone for the considered structure. The bold dashed curves represent the mode responsible for the refraction effects under analysis, specifically analyzed at the frequencies identified by the horizontal dash-dotted lines (see also Figs. 3 and 4).

the geometrical shape of the unit cell of the metamaterial, results in either types of ordinary and negative refraction, namely ordinary refraction for a positive angle incidence (red arrows) and negative refraction for a negative angle incidence (blue arrows).

The previously described phenomenology can be obtained by patterning one-half of the slab with a square array of L-shaped holes with a periodicity of 1125 nm (Fig. 1a, b and Supplementary Fig. 2). This minimal chiral shape (see also Fig. 2b) has already shown significant asymmetries and dichroism in the optical domain[39,40]; interestingly, it also leads to an asymmetric Bloch mode structure for mechanical waves propagating in the plane of the metasurface. Figure 1e reports the calculated mode dispersion for the patterned slab along high symmetry direction of the first Brillouin zone; details on the simulations can be found in the "Methods" section. The mode relevant for the asymmetric negative refraction, extending approximately in the 0.6–1.2 GHz frequency range, is depicted using bold, dashed lines.

## Isofrequency curves mapping

Our rationale was to experimentally characterize the isofrequency contours of the L-hole-shape metamaterial, to support the aforementioned prediction of asymmetric negative refraction. For this study we fabricated samples following the sketch in Fig. 2a. In order to excite the mechanical waves, we exploited the piezoelectric properties of GaAs[41] by directly fabricating interdigitated transducers (IDTs) on the device top surface. Applying a radio frequency voltage to the IDT, SAWs can be launched within the emission band determined by the periodicity of the IDT finger grating [Fig. 2b, d and Supplementary Fig. 3]. The SAW

generated by the IDT impinge on the metamaterial region and excites Bloch modes in it.

The acoustic wave field over the structure was mapped using the scanning optical interferometer sketched in Fig. 2d (see "Methods" and ref. 41 for details). This apparatus is designed to stroboscopically detect the spatially-resolved vertical instantaneous surface displacement, $w(x,y)$, with a spatial resolution of ~0.7 μm. A map for $w(x,y)$ recorded on a typical device excited at a frequency of 1.024 GHz is displayed in Fig. 2c. From the bottom to the top, one first recognizes the planar wavefronts of the SAW in the bulk GaAs region (i) between the IDT and the boundary of the metamaterial slab. The amplitude of $w$ in region (i) is estimated (see "Methods") to be of the order of 0.2–0.3 nm. Due to acoustic reflections at the metamaterial slab boundary, the SAW field in region (i) is expected to have a standing wave component. The latter excites acoustic modes within the metamaterial slab [region (ii)]: here, one observes a complex pattern consisting of modes with different propagation directions. Finally, a weak plane wave is subsequently transmitted to the bulk GaAs region beyond the membrane [region (iii)]. Since the acoustic aperture of the IDT is only slightly wider than the lateral size of the metasurface, significant diffraction phenomena occur at the corners of the patterned region (iv). In addition, both the underetched interface at the air spacer edges, and the metamaterial holes' sidewalls, have a non-negligible amount of roughness (see Supplementary Fig. 4). These features induce a deviation from the perfect periodical translational invariance along $\hat{x}$, and allow for wave scattering events that do not conserve the Bloch wavevector parallel to the unpatterned/patterned regions interface. Although size fluctuations are known to affect the performance of asymmetric metamaterials[42], in our system the scattering does not appreciably modify the Bloch mode dispersion, as it resulted a posteriori from the experiment-theory comparison (Fig. 3). Rather than being detrimental, scattering mechanisms are instead essential for the experimental determination of the 2D spatial dispersion of the metasurface (see the discussion about Fig. 3).

From the interferometric maps of instantaneous displacement, the corresponding reciprocal space maps can be obtained by applying a 2D Fourier transform; as detailed below, these maps are intimately connected to the isofrequency contours. Reciprocal space maps obtained experimentally at different frequencies for the metasurface shown in Fig. 2b are reported in the left panels of Fig. 3. As anticipated, even if the incident SAW has a wavevector oriented along $\hat{y}$ (i.e., $k_x = 0$; green markers), a non-negligible signal is found in patterns within the whole 2D reciprocal space, confirming that the SAW is subjected to scattering events coupling to many different wavevectors in the $k_x$–$k_y$ plane. Note that the maps of Fig. 3 are limited to half the reciprocal space (including only the positive $k_y$); this is due to the symmetry property of the modulus of the Fourier transform of a real quantity, which possesses point symmetry around Γ.

The $z$-displacement within the periodically patterned metasurface for a fixed excitation frequency, $w(x, y) \equiv w(\mathbf{r})$, can be theoretically written as a linear combination of Bloch's functions:

$$w(\mathbf{r}) = \sum_n \int_{\mathbf{k} \in K_n} d\mathbf{k}\, c_n(\mathbf{k})\, W_n(\mathbf{k}; \mathbf{r})\, e^{i\mathbf{k}\cdot\mathbf{r}} \qquad (2)$$

where $n$ is the mode index, $\mathbf{k}$ is the in-plane Bloch wavevector, $K_n$ is the isofrequency contour of the $n$-th mode (a one-dimensional subset of the first Brillouin zone), $c_n(\mathbf{k})$ are the linear combination weight coefficients, and $W$ is the periodic part of the Bloch's function. With some algebraic considerations, the Fourier transform (FT) of Eq. (2) can be written as:

$$\widetilde{w}(\mathbf{q}) = \sum_n \sum_{\mathbf{k} \in K_n} \sum_{\mathbf{g} \in RL} c_n(\mathbf{k})\, \widetilde{W}_n(\mathbf{k}; \mathbf{g})\, \delta(\mathbf{k} + \mathbf{g} - \mathbf{q}), \qquad (3)$$

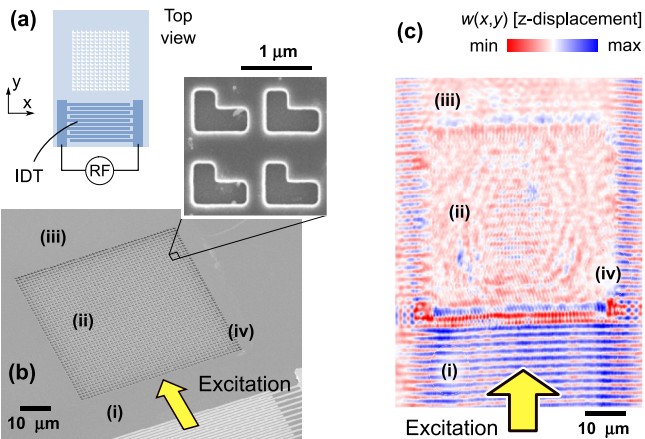

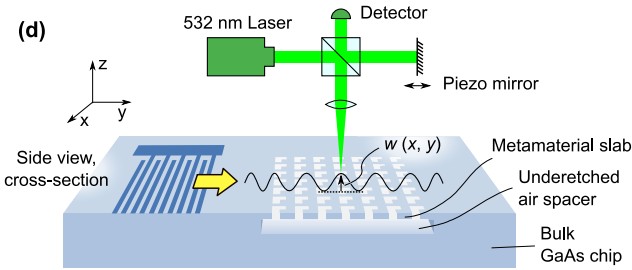

**Fig. 2 | Sample structure and interferometric readout of the mechanical motion. a** Sketch of sample structure. An interdigitated transducer (IDT) faces a patterned, suspended GaAs metamaterial slab. **b** SEM micrograph of a sample, showing a detail of the L-shape pattern. Labels (i–iv) indicate different regions detailed in the main text. **c** Typical measurement of instantaneous displacement. The excitation SAW is launched along the yellow arrow. **d** Sample cross-section and schematic of the interferometric apparatus employed to characterize the mechanical waves.

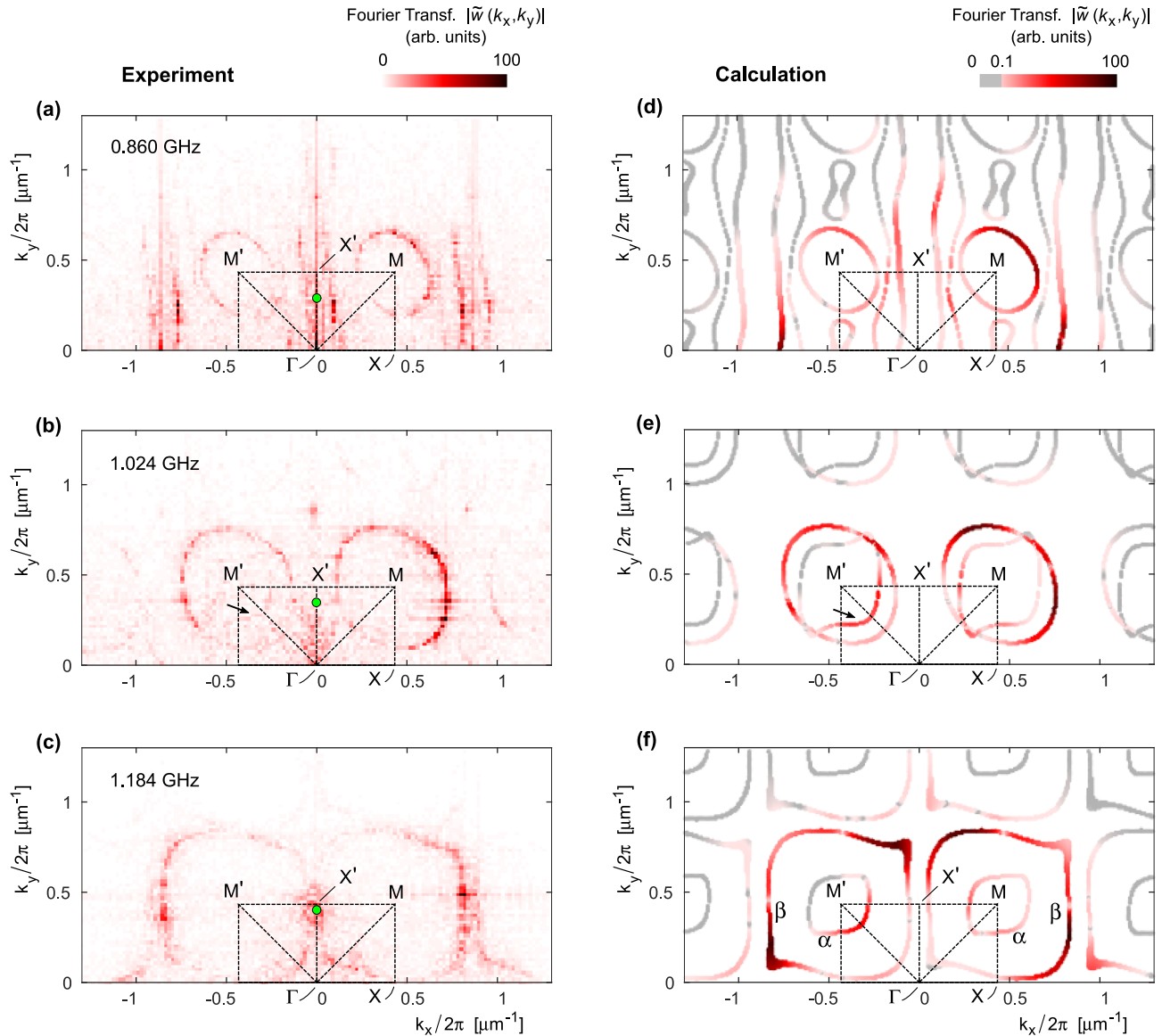

**Fig. 3 | Isofrequency curves in the reciprocal space.** Experimental (**a**–**c**) and simulated (**d**–**f**) reciprocal space maps for three different excitation frequencies, evidencing the Bloch mode isofrequency curves. The color encodes the modulus of the Fourier transform of out-of-plane mechanical displacement. The variations of such values (along a specific isofrequency curve, and among different curves) originate from two mechanisms: (i) intrinsic values of out-of-plane displacement for the different eigenstates, (ii) unequal coupling efficiency between the

excitation wave and the Bloch mode. Experimental maps are affected by both mechanisms, while calculated maps take into account only the first. The mode identified by β in panel (**f**) is that responsible for the anomalous refraction phenomena of interest in the present work (see also Fig. 4). The green markers in **a**–**c** represent the wavevector of the excitation SAW. The arrow in panel **e** indicates a portion of a strongly asymmetric isofrequency contour, that is also visible, yet very faintly, in the experimental data (see the arrow in **b**).

where the sum over **g** runs over the whole reciprocal lattice (RL). As can be seen, the FT of the displacement is proportional to two terms: the weights $c_n(\mathbf{k})$ described earlier, and the factor $\widetilde{W}$, that is the spatial FT of the function $W$ introduced in (2). Notice that the sum over **g** and the Dirac delta-factor implies that the $\widetilde{w}(\mathbf{q})$ is a "tiling" over the whole **q** space of the isosurfaces defined in the first Brillouin zone.

To calculate $\widetilde{w}(\mathbf{q})$ we resorted to the Finite Element Method (FEM —see "Methods" for details), from which $\widetilde{W}$ and $K_n$ can be directly calculated. An ab initio determination of the coefficients $c_n(\mathbf{k})$ is instead out of reach, since it would involve knowledge of the exact coupling mechanisms between the impinging SAW and the metamaterial Bloch modes, that in turn depend on the full geometry, including unintentional inhomogeneities of the bulk-membrane interface and of the hole sidewalls. We thus assumed $c_n(\mathbf{k}) = 1$ for every $n$ and **k**, that is, a simplifying assumption whose impact will be discussed in the

following. In addition, we convoluted the $\widetilde{w}(\mathbf{q})$ calculated with Eq. (3) with a suitable Gaussian function, describing the effect of the finite measuring laser spot size.

The calculated $\widetilde{w}(\mathbf{q})$, color-coded according to its modulus, is reported in the right panels of Fig. 3. Globally, the shapes of experimental and calculated curves are in excellent agreement. Looking instead at the color levels, the result is variable and depends on the regions considered: certain modes show a good agreement whereas other modes are weakly reproduced. We ascribe this fact to the assumption made on the $c_n(\mathbf{k})$: we can thus say a posteriori that not all the Bloch states belonging to the isofrequency curves are equally populated in the experiment. This effect is indeed reasonable: in a patterned slab like our metamaterial, Bloch modes are inherited from Lamb waves (with in-plane and out-of-plane motion) and from shear-horizontal waves (SH, with only in-plane motion). Since the excitation

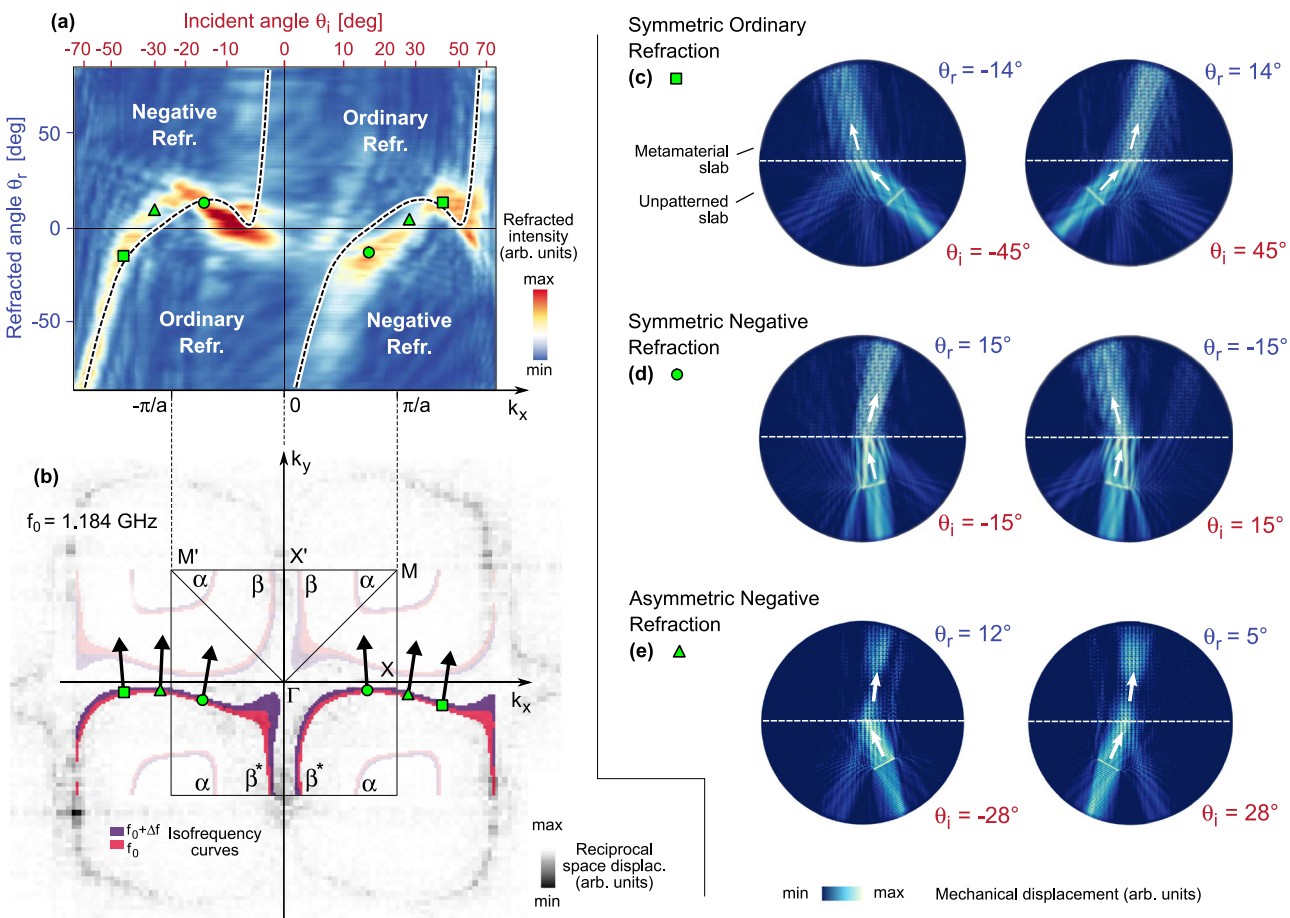

**Fig. 4 | Refraction characteristics for the interface between an ordinary slab and the asymmetric phononic metamaterial. a** Color map of the refracted wave intensity as a function of the incident and refracted angles, as calculated with the full-domain 3Dd finite-element model. The black dashed line represents the refraction relation determined from the gradient of the isofrequency curves. From the colormap, negative refraction (i.e., $\text{sign}(\theta_i) = -\text{sign}(\theta_r)$) occurs for $-40° < \theta_i < -8°$ and for $0° < \theta_i < 25°$; moreover, the map is not invariant under the exchange of $(\theta_i, \theta_r)$ with $(-\theta_i, -\theta_r)$. These observations indicate the existence of asymmetric negative refraction, that happens at the angle pair identified by triangular markers.

In the graph, each incident angle corresponds to a certain $k_x$ wavevector component, as provided by the double horizontal axis. Panel **b** reports as colored traces the calculated isofrequency curves superimposed with the measured metamaterial reciprocal space displacement map (grayscale). The data are the same as in Fig. 3c–f, but a symmetric domain over $k_y$ has been here employed; moreover, only the relevant sections of mode β (labeled as β*) are here highlighted. **c–e** Color maps illustrating the refracted beams in the various regimes enabled by the asymmetric metamaterial. The corresponding points in panels **a**, **b** are identified as green markers.

employed here is a $\hat{y}$-propagating Rayleigh SAW, with displacement only along $\hat{y}$ and $\hat{z}$, it is plausible that Lamb-like and SH-like Bloch modes are excited differently.

The reciprocal space maps reveal several peculiar features. At 0.86 GHz excitation (Fig. 3a, d) we found slightly asymmetric ellipsoidal isofrequencies around M and M' along with an open isofrequency contour extending vertically from Γ to X' and more. An open contour in the phase space can be linked to topological effects such as hyperbolic dispersion and Lifshitz transition, which have been recently reported in electromagnetic metamaterials[43,44]. At 1.024 GHz (Fig. 3b, e), the features become more localized with pairs of concentric, closed iso-contours. The isofrequency lines are centered around M and M', determining net directional band gaps around $k_x = 0$ and $k_y = 0$. The SAW direct excitation along Γ-X' gets quickly attenuated due to the absence of modes, thus favoring the population via scattering of modes at large $(k_x, k_y)$. Observing the color-code of the isolines, i.e., looking at the level of modal displacement, reveals another important effect. While the outer isofrequency curves clearly appear in the experimental map, the inner ones give a fainter signal, which can be recognized in the region indicated by the arrow. The position of the weak experimental feature matches with that of the (stronger) theoretical one, also indicated by an arrow. This effect can be understood in terms of mode matching: the inner isoline mode has a strong SH-like character (see Supplementary Fig. 5), which is probably excited with low efficiency by the SAW. This effect becomes even stronger at larger frequencies, namely 1.184 GHz, where the inner isofrequencies around M and M' (mode α) are invisible in the experiment. At this frequency, one finds another, larger isofrequency curve centered at M and M' (mode β); its structure is such that the directional band gap is strongly reduced. These isolines are very close to the Γ-X' segment, hence it is legitimate to expect a stronger population of a portion of the mechanical mode β from the ideally $k_x$-conserving SAW-Bloch mode scattering process. This can be indeed observed in the experiment as a hot spot close to the X' point. Most importantly, the isofrequency curve of mode β possesses a strong degree of asymmetry: it is exactly this feature that enables the special refraction phenomena described earlier.

## Emergence of asymmetric negative refraction

To illustrate the possible scenarios enabled by the metamaterial operating at 1.184 GHz, we started by determining the polar angle of the group velocity for mode β. In Fig. 4b, we report in pink/violet colors the calculated isofrequency curves, superimposed with the experimental Fourier displacement map (grayscale). We are here recalling the data of Fig. 3c, f, shown however in a different $k_y$ range—

that relevant for the present analysis. Moreover, the $k_y < 0$ section of calculated mode β are highlighted (and labeled as β*), since we are interested in group velocities $\mathbf{v}_g$ with a positive $\hat{\mathbf{y}}$-component (as a consequence of the conservation of energy flow and of the assumption that the incident wave is propagating upwards from the unpatterned slab half-space, see Fig. 1a). As illustrated in Fig. 4b, we can associate every point along the isofrequency contours to a group velocity vector. The angle $\theta_r$ that $\mathbf{v}_g$ forms with the $\hat{\mathbf{y}}$ direction has been plotted, as a function of $k_x$ and incident angle $\theta_i$, as black dashed lines in Fig. 4a (see Supplementary Fig. 1 for the relation between $k_x$ and $\theta_i$). Interestingly, the obtained trend shows the presence of ordinary and negative refraction, together with refraction asymmetry—as the two branches do not have the origin as symmetry center.

The results illustrated above leverage on the matching between the experimental results for the available Bloch states in the reciprocal space, and single-unit-cell mechanical simulations that yielded the isofrequency contours. To further push our analysis, we moved to a device-realistic FEM simulation, considering a circular domain, one-half of which is the unpatterned slab, and the other half is a macroscopic portion of the metamaterial (-1000 unit cells). A line source, with length smaller than the transversal extension of the metamaterial-unpatterned slab interface, excites antisymmetric Lamb waves in the unpatterned slab with a prescribed incidence angle $\theta_i$ (more details in the "Methods" section). The wave impinges on the interface, where refraction occurs (Fig. 4c–e). By evaluating the transmitted wave intensity in polar coordinates, we have retrieved the refracted intensity at angle $\theta_r$ upon excitation at angle $\theta_i$. Gathering data for all the ($\theta_i$, $\theta_r$) pairs, the color map (blue-white-yellow-red) of Fig. 4a has been determined. As it can be seen, the full-device FEM simulation (color map) mostly confirms the trend obtained by means of the group velocity extraction from the isofrequency contour (black dashed lines). The slight deviations between the red areas of the colormap and the black dashed lines could be attributed to the improved accuracy of the full-device FEM simulation, capable to capture also additional effects such as impedance-matching. For instance, the isofrequency contour model predicts that, for an incident wave with $|\theta_i|$ getting close to 0, the refracted wave has a strongly growing $|\theta_r|$ (i.e., the refracted wave has grazing propagation); this effect is due to the shape of the isofrequency contours, that, when $k_x$ is getting close to 0, are almost parallel to the $k_y$ axis. The resulting refraction branches with strongly growing $|\theta_r|$ are only weakly observed in the full-device FEM model: we indeed attribute this difference to impedance mismatch. From now on, we will focus upon the full FEM model (color map), which is expected to represent more closely the behavior of real devices.

By selecting the angle of incidence, it is possible to enable very different refraction regimes for the mechanical waves. For instance, when $|\theta_i|$ - 45°, we have *symmetric ordinary refraction*: a wave incident at 45° gets refracted at 14° while a wave incident at −45° gets refracted at −14°; this is illustrated by square markers in Fig. 4a and by the displacement maps of Fig. 4c. A simple tuning of incident angle allows access to negative refraction (NR, i.e., sign($\theta_i$) = −sign($\theta_r$)), that occurs in the second and fourth quadrants of Fig. 4a. In particular, *symmetric negative refraction* occurs choosing $\theta_i = \pm15°$, since at 15° (−15°) incidence corresponds a −15° (15°) refraction, see the circle markers in Fig. 4a and the maps of Fig. 4d. Our metamaterial can, however, enable the most peculiar case, that is *asymmetric negative refraction* (triangle markers in Fig. 4a, e) for an incident angle of magnitude equal to 28°. If such angle is chosen to be positive (+28°) the refracted angle is also positive (+5°); however, if the incident angle is set to be negative (−28°), the refracted wave occurs still at a positive angle, yet different from the previous one (+12°).

## Discussion

Although propagation of waves in linear media is a seemingly simple and well-understood topic, the analysis of systems where the local

response function (compliance tensor and mass density for mechanical waves, permittivity and permeability for electromagnetic waves...) is spatially inhomogeneous deserves research efforts, as it allows for very peculiar propagation and transport phenomena. Under this rationale, we have demonstrated that a quite simple system, that is a mechanically compliant plate patterned over a half-space with asymmetric L-shaped holes, manifests a phenomenon that is strongly unusual and—to the best of our knowledge—has never been reported so far: asymmetric negative refraction (ANR). Such effect follows from the existence of mechanical Bloch modes—and concurrent isofrequency curves—of minimal symmetry in the first Brillouin zone. When such modes are evaluated at specific pairs of points in the Brillouin zone (i.e., at the $k_x$ and $-k_x$ highlighted by green triangles in Fig. 4b), the $\hat{\mathbf{x}}$-component of the group velocities $\mathbf{v}_g$ remains of the same sign. While our model does not rely on an analysis of the metamaterial constitutive relations, but rather on a direct calculation of the Bloch modes, further studies could focus on the possible role of Willis-type coupling, and of the concurrent propagation asymmetries, occurring on the L-holed plate[45,46].

Our proof-of-principle report relies on an extremely simple metamaterial design, the L-shaped hole array on a square lattice; it has yielded clear evidence of the ANR phenomenon, yet in a narrow incident angular range and with access to small refracted angles. Making reference to recent results in electromagnetic metamaterial optimization[47], it is however reasonable to expect that other holes' shapes and distribution would allow for more pronounced and robust operation, keeping at the same time the multifunctional character of the device.

Beyond the foundational character of ANR, whose possible connection with advanced wave physics concepts such as topological invariants are still to be elucidated, and potentially impacting fields as diverse as seismic engineering and nanophotonics[48,49], one could appreciate the possible application perspectives of a device where diverse refraction regimes are concurrently present, and operating at GHz frequency in a piezoelectric semiconductor. Those include, in a non-exhaustive list, multiplexed radio-frequency filters, multi-channel optical modulators and modulated (quantum) light emitters, phononic cloaks, and shields, where further functionalities can be included thanks to the possible exploitation of material platforms other than monolithic gallium arsenide (for instance, aluminum nitride on silicon), or by heterogeneous integration techniques.

## Methods
### Simulations
The calculations presented in the article have been performed relying on the Finite Element Modeling technique (FEM), utilizing the commercial software COMSOL Multiphysics. Two different kind of simulations have been run: Bloch mode calculations (Figs. 1e, 3d–f, 4b) and wave refraction calculations (Fig. 4c–e). In Bloch mode calculations, the simulation domain consisted of a single unit cell of the metamaterial, with Bloch-Floquet boundary conditions for the xz- and yz-boundary planes; the GaAs-air boundary was modeled as a free boundary (i.e., no air-induced damping; at these frequencies, the presence of air would likely affect the resonance Q by a factor two[50]). Bloch modes were first calculated with the eigenfrequency solver by parametrically sweeping the in-plane Bloch wavevector, and the isofrequency contours have been subsequently extracted by search and array indexing (in a post-processing MATLAB script). In refraction calculations, a much larger FEM computational domain has been used. It consisted of a cylinder of thickness 210 nm and of radius 45.6 μm, including an outer perfectly matched layer (annular-shaped, of depth 17.5 μm) region and -1000 unit cells of the L-hole metamaterial, representing the half-patterned slab (Figs. 1a and 4c–e). Waves were excited by a time-harmonic line displacement source and the solution found in the frequency domain.

In both Bloch mode and refraction calculations, we solved the solid mechanics linear wave equation for a piezoelectric medium, using the following parameters: $c_{11} = c_{22} = c_{33} = 11.88 \cdot 10^{10}$ Pa, $c_{12} = c_{13} = c_{23} = 5.38 \cdot 10^{10}$ Pa, $c_{44} = c_{55} = c_{66} = 5.94 \cdot 10^{10}$ Pa; $e_{14} = e_{25} = e_{36} = 0.154$ C/m$^2$; $\varepsilon = 12.46$; $\rho = 5320$ kg/m$^2$, where $c_{IJ}$ and $e_{iJ}$ are, respectively, the stiffness and piezoelectric tensor elements (in Voigt notation and stress-charge form), $\varepsilon$ is the constant-strain relative permittivity, and $\rho$ is the density. Note that the tensors are given with respect to the cubic conventional cell axes of GaAs; since the meta-surface L-shape hole has its sides (i.e., $x$ and $y$ axes in Fig. 1) parallel to the [110] and [−110] crystalline directions, a tensor rotation (whose implementation is available in COMSOL) was required. A 2% rescaling on the frequencies was applied to find exact matching with the experiments; this can be ascribed to inaccuracies in the employed values for the material parameters, that have been retrieved from databases and not directly measured on our GaAs thin film.

### Fabrication

Sample fabrication begins with molecular beam epitaxy of $Al_{0.5}Ga_{0.5}As$ (500 nm) and GaAs (210 nm) layers on an undoped GaAs wafer. Subsequently, the L-shape hole pattern is defined by means of e-beam lithography (Zeiss Ultraplus SEM + Raith Multibeam system) on AR 6200 resist (AllResist) and ICP-RIE (Sentech), using $BCl_3$-$Cl_2$-Ar gas mixture. Interdigitated transducers (IDTs) are then defined by e-beam lithography (AR-P 679.04 PMMA resist by AllResist), metal (Au, 50 nm) thermal evaporation, and lift-off. AC lines and bonding pads are realized by optical lithography (ML3 laser writer by Durham Magneto-Optics) and metal (Cr/Au, 7/80 nm) thermal evaporation. The meta-material membrane is underetched by a hydrofluoric acid bath and dried in a critical point dryer equipment (Tousimis). Finally, the sample, that features an array of different-pitch IDTs, each of which facing a metamaterial membrane, is wire-bonded to a PCB that hosts RF lines and connectors.

### Interferometric displacement mapping

The SAW mappings were recorded using the scanning Michelson interferometer depicted in Fig. 2d. The interferometer operates with a single mode laser with a wavelength $\lambda_L = 532$ nm focused by a ×50 microscope objective (Mitutoyo M Plan NIR) onto a ~0.7 µm spot on the sample surface. The sample is mounted on a $xy$-piezoelectric scanning table with a positioning resolution of 50 nm. The mirror at the reference arm of the interferometer is mounted on a piezo translation stage whose position is actively controlled by a feedback loop. The latter maintains a constant optical phase difference of $\lambda_L/4$ between the two arms during scanning, which ensures a maximum sensitivity to the excitation-induced surface displacement $w$. The optical beam exiting the beam splitter is detected by a high-frequency detector connected to a spectrum analyzer. All the experiments have been performed at room-temperature and atmospheric pressure.

The IDTs on the chip were powered with a RF generator (20 dBm nominal power, i.e., not corrected for losses in the RF cables and on the PCB). Typical RF scattering parameter $S_{11}$ (corresponding to the electrical power reflection) have dips in correspondence of the IDT resonance frequency with depths of 0.5–1.5 dB. By neglecting losses in the connections, the linear acoustic power density is estimated to be 12–48 W/m. By comparing with a numerical model, this yields a surface displacement amplitude in the range 0.26–0.6 nanometers outside the metamaterial. Interferometric profiles along the SAW propagation path (Fig. 2) show that the field displacements inside the metamaterial are about one order of magnitude smaller, but always above the noise floor of the instrument.

### Data availability

Raw data, data analysis, and plotting scripts are available at https://doi.org/10.5281/zenodo.7016872.

### Code availability

Computer codes for finite-element calculations are available at https://doi.org/10.5281/zenodo.7016872.

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

## Acknowledgements

We acknowledge funding from ATTRACT, a research and innovation project funded by EU, GA 101004462; A.P. acknowledges funding from Alexander von Humboldt Foundation through the Experienced Researcher Fellowship program. We thank A. Tredicucci for a critical reading of the manuscript.

## Author contributions

S.Z. has conceived the experiment and fabricated the samples, starting from materials grown by G.B. P.V.S. and S.Z. have performed the measurements. A.P. has developed the numerical model. All the authors have contributed to the data analysis and to manuscript writing.

## Competing interests

The authors declare no competing interests.
