## [Peer Review File · Nature Communications]

Metamaterial-enabled asymmetric negative refraction of GHz mechanical wavesREVIEWER COMMENTS

Reviewer #1 (Remarks to the Author):

Negative refraction is widely investigated in wave metamaterials, while the asymmetry between the incident angles is rarely explored before. The authors experimentally demonstrated the novel negative refraction operating at GHz frequency with a chip-scale metamaterial design. With the asymmetric sub-wavelength components, the shape of the Bloch mode isofrequency curves is judiciously modulated, which gives us more freedom to engineer the general Snell's law at interfaces. The manuscript is well written and the experimental results are rather solid. I believe that this work may stimulate some immediate applications in wave fields such as optomechanics, and recommend for publication after proper revisions:

1)Acoustic Willis metamaterials also have asymmetric responses for different incident angles. The connections and differences should be shortly discussed in the introduction.

2)Part of the measured dispersion in Fig 4(a) are missing, such as the cases around -50 degrees and $k_x > 0$. Also, the measured curves are not perfectly consistent with the theory. Please add some clear explanations.

3)On Page 9 Line 28, the authors claimed that mode β possesses a strong degree of asymmetry: it is exactly this feature that will enable the special refraction phenomena. Can you add some quantitative discussion about the asymmetry property?

4)How to determine maximum sensitivity of the excitation-induced surface displacement?

5)The authors present some nice experimental results for the cases of oblique incidence with positive and negative angles. A direct question is: how the waves will be refracted for the vertical incidence in this design? Please also give some general discussions for similar systems.

Reviewer #2 (Remarks to the Author):

The manuscript reports on an experimental realization of asymmetric negative refraction in a metasurface. The experimental work and the relevant data analysis is done very carefully, resulting in the good match between extracted and calculated isofrequency maps of Fig 3. The section on isofrequency curve mapping is well written, with an appropriate level of details. Similarly, the Methods are written appropriately. Overall, the objectives are clear and the results support the findings. I would have preferred to see this work in PRB, but I acknowledge that Nature Communications is also an appropriate venue. I have a few suggestions for minor improvements of the manuscript to make it acceptable for publication.

The effect of imperfections in the manufacturing process on the wave field needs to be discussed in more detail. Lines 31-32 of the supplementary allude to this. Comment on the robustness of the observed phenomenon to deviations from periodicity.

Many of the sentences are very long and sometimes twisted. This makes reading burdensome. Please revise the manuscript to address this. A more readable paper will be better received.

Some of the terminology and expressions used in the text are unsupported or unnecessarily complicated. "Subwavelength structuration" on line 13 is very odd. I recommend against creating new words; the field of metamaterials is already riddled with unnecessary terms. I do not find "Possibly operating in the quantum regime" on line 20 very well supported by this work; the expression is thus not suitable for the abstract. On line 51, "perhaps this particular configuration is ill-suited for integration with other on-chip technologies", the claim is left unsupported; clarify. On line 285, "an educated guess allows however to state that" can be revised or even completely omitted; this 5-line sentence can benefit from revision.

Reviewer #3 (Remarks to the Author):

The article outlines an experimentally feasible approach to achieve anomalous refraction of mechanical waves at an interface in the GHz range through the use of a micron scale square array of asymmetric holes. While the ability to achieve exotic refraction through the use of metamaterials with broken spatial symmetries is not new, this paper presents a clear elucidation of how such an effect can be achieved at a remarkably high acoustic frequency in the context of Lamb-like waves in a lithographically patterned GaAs plate.

The asymmetric band structure is experimentally measured with SAW excitation and shows remarkably good agreement with numerical simulations. It is this band structure that enables the asymmetric negative refraction. This refraction, while convincingly demonstrated in finite element simulations, is however not shown experimentally. The authors could do more to provide context for their results. For example, while some literature on high frequency acoustic metamaterials is discussed, the focus is on topological phenomena. While this work certainly points to possible realization of hypersonic time-reversal invariant topological acoustics (such as that achieved in Q. Zhang, et al. Nat Electron 5, 157–163 (2022)), it would seem more relevant to highlight the literature on high frequency phononics at large and, in particular, any work done in high frequency acoustic refraction (see, for example, review articles such as M. Maldovan, Nature 503, 209–217 (2013); T. Vasileiadis, et al., J Appl Phys 129, 160901 (2021); M. Nomura, et al., APL Materials 10, 050401 (2022); and C. Choi, et al., Adv. Eng. Mater., 23: 2000988, (2021)).

The data analysis and interpretations are excellent. The narrative provided in the text, supported by high quality figures, make the article convincing, accessible to a broad audience, and enjoyable to read. Enough detail is provided to replicate all results.

Minor comments

- A discussion about the effect of air induced damping would be appreciated. The article states that it is neglected in the simulations, but this should be justified.
- The title could be changed to emphasize that the paper's contribution is asymmetric negative refraction in a micron scale acoustic platform (the theoretical basis for general metamaterial-enabled refraction effects such as asymmetric negative refraction is already well-understood).
- Figs 1c and 1d: the frequency f_0 for these results should be specified.
- Fig 1: The caption for (b) should come before (c, d).
- Figs 3b and 2e: the black arrows should be explained or removed.
- Fig 4: recommend that a colorbar be included in the figure as opposed to specifying it in the caption. A colorbar could also be helpful for the color plots in (c), (d), and (e).
- It could be helpful to include a little more description about the "array of different-pitch IDTs" such as how they were arranged and the considerations taken so that the three different frequencies could be generated, whether on a single sample or on different samples.

Best,
Curtis Rasmussen

Reviewer #1 (Remarks to the Author):

Negative refraction is widely investigated in wave metamaterials, while the asymmetry between the incident angles is rarely explored before. The authors experimentally demonstrated the novel negative refraction operating at GHz frequency with a chip-scale metamaterial design. With the asymmetric sub-wavelength components, the shape of the Bloch mode isofrequency curves is judiciously modulated, which gives us more freedom to engineer the general Snell's law at interfaces. The manuscript is well written and the experimental results are rather solid. I believe that this work may stimulate some immediate applications in wave fields such as optomechanics, and recommend for publication after proper revisions:

1) Acoustic Willis metamaterials also have asymmetric responses for different incident angles. The connections and differences should be shortly discussed in the introduction.

We acknowledge that we have missed any mention of Willis metamaterials, that could indeed be related to our system. We have inserted a sentence in Pag. 12, and Refs. 45,46: addressing this point:

While our model does not rely on an analysis of the metamaterial constitutive relations, but rather on a direct calculation of the Bloch modes, further studies could focus on the possible role of Willis-type coupling, and of the concurrent propagation asymmetries, occurring on the L-holed plate^{45,46}.

2) Part of the measured dispersion in Fig 4(a) are missing, such as the cases around -50 degrees and $k_x > 0$. Also, the measured curves are not perfectly consistent with the theory. Please add some clear explanations.

In Pag. 10 we have modified and expanded a sentence, clarifying possible reasons to the partial data inconsistency.

As it can be seen, apart for a slight shift, the full device FEM simulation (color map) confirms the trend obtained by means of the group velocity extraction from the isofrequency contour (black dashed lines).

→ As it can be seen, the full-device FEM simulation (color map) mostly confirms the trend obtained by means of the group velocity extraction from the isofrequency contour (black dashed lines). The slight deviations between the red areas of the colormap and the black dashed lines could be attributed to the improved accuracy of the full-device FEM simulation, capable to capture also impedance-matching effects. Notice indeed that, getting close to $k_x = 0$ (i.e. vertical incidence), the refracted wave has a grazing propagation angle.

3) On Page 9 Line 28, the authors claimed that mode β possesses a strong degree of asymmetry: it is exactly this feature that will enable the special refraction phenomena. Can you add some quantitative discussion about the asymmetry property?

The Reviewer's comment made us aware that the sentence was unclear, since we were referring to the isofrequency curve of mode β and not on the field mode itself. Indeed, the quantitative discussion about the isofrequency curve of mode β was already analyzed in depth around Fig. 4. We have modified the sentence in Pag. 9, first paragraph, to clarify this point:

Most importantly, mode β possesses a strong degree of asymmetry: it is exactly this feature that will enable the special refraction phenomena described earlier.

→ Most importantly, the isofrequency curve of mode β possesses a strong degree of asymmetry: it is exactly this feature that enables the special refraction phenomena described earlier.

4) How to determine maximum sensitivity of the excitation-induced surface displacement?

The interferometric measurements were not calibrated to yield the absolute amplitude of the SAW fields. One of the reasons was that the calibration is not straightforward since the PC membrane is partially transparent to the laser light, so that the interferometric amplitude becomes dependent not only on the surface reflection but also on the absorption coefficient as well as the reflectivity of the membrane and regions underneath it. We estimated the acoustic amplitudes using the alternative procedure already partially illustrated in the *Methods* section. We expanded that Section, clarifying that

“Interferometric profiles along the SAW propagation path (Fig. 2) show that the field displacements inside the metamaterial are about one order of magnitude smaller [than 0.26-0.6 nm], but always above the noise floor of the instrument.”

5) The authors present some nice experimental results for the cases of oblique incidence with positive and negative angles. A direct question is: how the waves will be refracted for the vertical incidence in this design? Please also give some general discussions for similar systems.

In our work we have analyzed only non-normal incidence because our metamaterial has a directional bandgap for waves propagating at normal incidence, as already mentioned in Pag 8 (“The isofrequency lines are centered around M and M’, determining net directional band gaps around $k_x=0$ and $k_y=0$ ”). In our revision we further stress on this point with the new sentence in Pag. 10:

Notice indeed that, getting close to $k_x = 0$ (i.e. vertical incidence), the refracted wave has a grazing propagation angle.

Reviewer #2 (Remarks to the Author):

The manuscript reports on an experimental realization of asymmetric negative refraction in a metasurface. The experimental work and the relevant data analysis is done very carefully, resulting in the good match between extracted and calculated isofrequency maps of Fig 3. The section on isofrequency curve mapping is well written, with an appropriate level of details. Similarly, the Methods are written appropriately. Overall, the objectives are clear and the results support the findings. I would have preferred to see this work in PRB, but I acknowledge that Nature Communications is also an appropriate venue. I have a few suggestions for minor improvements of the manuscript to make it acceptable for publication.

The effect of imperfections in the manufacturing process on the wave field needs to be discussed in more detail. Lines 31-32 of the supplementary allude to this. Comment on the robustness of the observed phenomenon to deviations from periodicity.

As discussed in the SM, imperfections play a pivotal role for experimentally observing the maps reported in Fig. 3. Imperfection-based scattering can arise indeed from several “defects”, ranging from surface scattering, inhomogeneous hole shape and periodicity, boundary between suspended and not-suspended regions, etc. While identifying the role of each class of imperfections is out of the scope of our manuscript, we can observe that the matching between experiment and calculations was systematic; in addition to the 3 devices whose data are reported in the article, statistics include 4 more devices with different hole symmetries. To address this point we have expanded the sentence

Rather than being detrimental, scattering mechanisms are essential for the experimental determination of the 2D spatial dispersion of the metasurface (see the discussion about Fig. 3).

that now reads

Although size fluctuations are known to affect the performance of asymmetric metamaterials⁴², in our system the scattering does not appreciably modify the Bloch mode dispersion, as it resulted *a posteriori* from the experiment-theory comparison (Fig. 3). Rather than being detrimental, scattering mechanisms are instead essential for the experimental determination of the 2D spatial dispersion of the metasurface (see the discussion about Fig. 3).

We have also added Ref. 42 that is relevant in the context.

Many of the sentences are very long and sometimes twisted. This makes reading burdensome. Please revise the manuscript to address this. A more readable paper will be better received.

Some of the terminology and expressions used in the text are unsupported or unnecessarily complicated. “Subwavelength structuration” on line 13 is very odd. I recommend against creating new words; the field of metamaterials is already riddled with unnecessary terms. I do not find “Possibly operating in the quantum regime” on line 20 very well supported by this work; the expression is thus not suitable for the abstract. On line 51, “perhaps this particular configuration is ill-suited for integration with other on-chip technologies”, the claim is left unsupported; clarify. On line 285, “an educated guess allows however to state that” can be revised or even completely omitted; this 5-line sentence can benefit from revision.

We acknowledge the Reviewer for pointing out such weaknesses. To address that we performed the following modifications:

Pag 1:

~~Under specific circumstances, mostly enabled by subwavelength structuration [...]~~

→ Under specific circumstances, mostly enabled by structuration below the wavelength scale [...]

Pag 1:

~~Our study specialized upon a mechanical metamaterial operating at GHz frequency, which is by itself a building block for advanced technologies such as chip-scale hybrid optomechanical and electromechanical devices operating in the quantum regime.~~

→ Our study specialized upon a mechanical metamaterial operating at GHz frequency, which is by itself a building block for advanced technologies such as chip-scale hybrid optomechanical and electromechanical devices.

Pag 2:

~~It is worth mentioning that topological effects have been investigated in one-dimensional heterostructures up to 300 GHz [Ref. 26], yet this particular configuration is ill suited for integration with other on-chip technologies.~~

→ It is worth mentioning that topological effects have been investigated in one-dimensional heterostructures up to 300 GHz [Ref. 28], yet this particular, non-planar configuration is not ideal for integration with other on-chip technologies.

Pag 12:

~~Making reference to recent results in electromagnetic metamaterial optimization, an educated guess allows however to state that, upon appropriate reconfiguration of the holes' shapes and distribution, it is possible to obtain more pronounced and robust operation, keeping at the same time the multifunctional character of the device — i.e., the concurrent presence of symmetric ordinary refraction, symmetric negative refraction, and asymmetric negative refraction.~~

→ Making reference to recent results in electromagnetic metamaterial optimization, it is however reasonable to expect that other holes' shapes and distribution would allow for more pronounced and robust operation, keeping at the same time the multifunctional character of the device.

Reviewer #3 (Remarks to the Author):

The article outlines an experimentally feasible approach to achieve anomalous refraction of mechanical waves at an interface in the GHz range through the use of a micron scale square array of asymmetric holes. While the ability to achieve exotic refraction through the use of metamaterials with broken spatial symmetries is not new, this paper presents a clear elucidation of how such an effect can be achieved at a remarkably high acoustic frequency in the context of Lamb-like waves in a lithographically patterned GaAs plate.

The asymmetric band structure is experimentally measured with SAW excitation and shows remarkably good agreement with numerical simulations. It is this band structure that enables the asymmetric negative refraction. This refraction, while convincingly demonstrated in finite element simulations, is however not shown experimentally. The authors could do more to provide context for their results. For example, while some literature on high frequency acoustic metamaterials is discussed, the focus is on topological phenomena. While this work certainly points to possible realization of hypersonic time-reversal invariant topological acoustics (such as that achieved in Q. Zhang, et al. Nat Electron 5, 157–163 (2022)), it would seem more relevant to highlight the literature on high frequency phononics at large and, in particular, any work done in high frequency acoustic refraction (see, for example, review articles such as M. Maldovan, Nature 503, 209–217 (2013); T. Vasileiadis, et al., J Appl Phys 129, 160901 (2021); M. Nomura, et al., APL Materials 10, 050401 (2022); and C. Choi, et al., Adv. Eng. Mater., 23: 2000988, (2021)).

We acknowledge the Reviewer for providing the links to broad review articles that we did not include in the first version of our manuscript. We chose from the above list two recent papers that we believe to be the most appropriate for our context, and included in the revised manuscript as Ref. 11-12.

The data analysis and interpretations are excellent. The narrative provided in the text, supported by high quality figures, make the article convincing, accessible to a broad audience, and enjoyable to read. Enough detail is provided to replicate all results.

Minor comments

- A discussion about the effect of air induced damping would be appreciated. The article states that it is neglected in the simulations, but this should be justified.

In our original manuscript we neglected to discuss this point since the effect under investigation did not appear to require the introduction of air damping in the model. Anyway, we agree that air damping is in general an important effect; for this reason we modified a sentence, introducing an estimate of the magnitude of this effect in our system:

the GaAs-air boundary was modeled as a free boundary (i.e., no air-induced damping).

→ the GaAs-air boundary was modeled as a free boundary (i.e., no air-induced damping; at these frequencies, the presence of air would likely affect the resonance Q by a factor two [50]).

- The title could be changed to emphasize that the paper's contribution is asymmetric negative refraction in a micron scale acoustic platform (the theoretical basis for general metamaterial-enabled refraction effects such as asymmetric negative refraction is already well-understood).

We recognize that the original title was perhaps too broad. Our proposal for a new title is

"Metamaterial-enabled asymmetric negative refraction of GHz mechanical waves"

- Figs 1c and 1d: the frequency f_0 for these results should be specified.

The Reviewer's comment made us aware that in the original manuscript it was unclear that Fig. 1c-d are a qualitative sketch, that does not derive from an exact numerical calculation, and that we reported with the intent of visually clarifying the quantitative data reported instead in the other figures. We have modified Fig. 1 caption in the following way:

~~(c, d) – Sketch of the [...]~~

→ (c, d) – Qualitative sketch of the [...]

and a sentence in Pag. 3:

~~This effect can be understood by looking at the sketch of the isofrequency curves in the reciprocal space reported in (c), where the energy increasing trend of the isolines has been color-coded from pink to violet.~~

→ This effect can be understood by looking at the qualitative sketch of the isofrequency curves in the reciprocal space reported in (c), where the energy-increasing trend of the isolines has been color-coded from pink to violet. Here, f_0 is a frequency illustratively around 1.18 GHz; quantitative data are shown in Fig. 3.

- Fig 1: The caption for (b) should come before (c, d).

The caption of Fig. 1 in the revised manuscript now reports the lettering in correct order.

- Figs 3b and 2e: the black arrows should be explained or removed.

We introduced the arrows on purpose to highlight a faint feature, that was discussed in the main text. For greater clarity we now make reference to the arrows in Fig. 3 caption:

A faint feature in the experimental data, and its theoretical counterpart, is indicated by an arrow in (b) and (e).

- Fig 4: recommend that a colorbar be included in the figure as opposed to specifying it in the caption. A colorbar could also be helpful for the color plots in (c), (d), and (e).

We recognize that in general colorbars are more readable than verbal descriptions of the palette in a color plot. However we found difficulties in graphically fitting the colorbar in the layout of Fig. 4, hence we opted for the wording in the caption. For consistency reasons we maintained the same approach for the color plots in (c,d,e):

(c-e) color maps illustrating the refracted beams in the various regimes enabled by the asymmetric metamaterial (dark blue = low displacement, light blue = high displacement).

- It could be helpful to include a little more description about the “array of different-pitch IDTs” such as how they were arranged and the considerations taken so that the three different frequencies could be generated, whether on a single sample or on different samples.

To address the Reviewer request we prepared an additional Supplementary Figure with general details on the sample structure, sample mounting, and on the array of IDTs. This figure is now Supplementary Figure 3; the others have been renumbered accordingly.

REVIEWERS' COMMENTS

Reviewer #1 (Remarks to the Author):

The revised manuscript has almost fully addressed my questions and concerns. The experimental demonstration of the anomalous refraction, mentioned by the referee 3, has partly been verified by the band structures in the original draft. I agree that direct visualization of the anomalous refraction should be added to better display the characteristics. Besides, the grazing propagation angle needs further explanation and a colorbar should be added for Fig 4(a)(c).

Reviewer #2 (Remarks to the Author):

A new comment is added at the end of the caption for Figure 3 about the arrows in panels (b) and (e). This comment is not clarifying what the faint feature in the panels are. It is better if the authors clarify in the text what the faint feature is, why it appears, and what its significance is.

Overall, the response from the authors is rather terse and impatient. Nevertheless, the authors have adequately addressed my technical concerns.

Reviewer #3 (Remarks to the Author):

The authors have adequately addressed my concerns and I recommend that the article be published. The paper presents a convincing experimental demonstration of high-frequency acoustic metamaterial behavior and is well supported by numerical simulations. While I would have appreciated more insight in the discussion of how this work relates to other work in high-frequency acoustic metamaterials, my opinion is that the paper be published as is.

Reviewer #1 (Remarks to the Author):

The revised manuscript has almost fully addressed my questions and concerns. The experimental demonstration of the anomalous refraction, mentioned by the referee 3, has partly been verified by the band structures in the original draft. I agree that direct visualization of the anomalous refraction should be added to better display the characteristics. Besides, the grazing propagation angle needs further explanation and a colorbar should be added for Fig 4(a)(c).

We are glad to know that the Reviewer finds an almost complete addressing of their questions and concerns in the first revised version of the manuscript. Regarding the question that they raise about adding a “direct visualization of the anomalous refraction”, we think that Fig. 4e already fulfils this request. There the anomalous refraction is visualized in what we believe to be the most direct way, i.e., with mechanical displacement maps showing the spatial distribution of the refracted beams. For this reason, we do not deem it appropriate to introduce modifications to the current state of data display. Concerning the role of grazing propagation angle, we recognize that our discussion was too concise. We thus clarified the point expanding and rephrasing a sentence in Pag. 10:

Notice indeed that, getting close to $k_x=0$ (i.e. vertical incidence), the refracted wave has a grazing propagation angle.

For instance, the isofrequency contour model predicts that, for an incident wave with $|\theta_i|$ getting close to 0, the refracted wave has a strongly growing $|\theta_r|$ (i.e., the refracted wave has grazing propagation); this effect is due to the shape of the isofrequency contours, that, when k_x is getting close to 0, are almost parallel to the k_y axis. The resulting refraction branches with strongly growing $|\theta_r|$ are only weakly observed in the full-device FEM model: we indeed attribute this difference to impedance mismatch.

Finally, we have added colorbars to Fig. 4(a)(c), that greatly benefitted its clarity.

Reviewer #2 (Remarks to the Author):

A new comment is added at the end of the caption for Figure 3 about the arrows in panels (b) and (e). This comment is not clarifying what the faint feature in the panels are. It is better if the authors clarify in the text what the faint feature is, why it appears, and what its significance is.

Overall, the response from the authors is rather terse and impatient. Nevertheless, the authors have adequately addressed my technical concerns.

We are glad to understand the overall appreciation that the Reviewer has deserved to the technical content of our manuscript revision. We recognise that our description about the feature indicated by the arrow in Fig. 3 (b), (e) was too concise; hence, the present revision of the manuscript embeds a clearer phrasing of the caption to Fig. 3 and of a sentence in Pag. 9:

A faint feature in the experimental data, and its theoretical counterpart, is indicated by an arrow in (b) and (e).

→ The arrow in panel (e) indicates a portion of a strongly asymmetric isofrequency contour, that is also visible, yet very faintly, in the experimental data (see the arrow in b).

Regarding mode excitation, in accordance with the color code of the simulation, the outer isofrequency curves clearly appear in the experimental map, while the inner ones give a fainter signal, which can still be recognized in the region indicated by the arrow.

→ Observing of the color-code of the isolines, i.e. looking at the level of modal displacement, reveals another important effect. While the outer isofrequency curves clearly appear in the experimental map, the inner ones give a fainter signal, which can be recognized in the region indicated by the arrow. The position of the weak experimental feature matches with that of the (stronger) theoretical one, also indicated by an arrow. This effect can be understood in terms of mode matching: the inner isoline mode has a strong SH-like character (see Supplementary Fig. 5), which is probably excited with low efficiency by the SAW.

Reviewer #3 (Remarks to the Author):

The authors have adequately addressed my concerns and I recommend that the article be published. The paper presents a convincing experimental demonstration of high-frequency acoustic metamaterial behavior and is well supported by numerical simulations. While I would have appreciated more insight in the discussion of how this work relates to other work in high-frequency acoustic metamaterials, my opinion is that the paper be published as is.

We are glad to know of the recommendation by Reviewer #3 and of their appreciation of our work. We are sorry not having addressed in full their request to provide a deeper contextualisation of our work within the literature of high-frequency acoustic metamaterials; however, we believe that this would have expanded too much the introduction at the expense of the other sections of our research article.